# Lipase Inactivation Kinetics of Tef Flour with Microwave Radiation and Impact on the Rheological Properties of the Gels Made from Treated Flour

**DOI:** 10.3390/molecules28052298

**Published:** 2023-03-01

**Authors:** Workineh Abebe, Grazielle Náthia-Neves, Caleb S. Calix-Rivera, Marina Villanueva, Felicidad Ronda

**Affiliations:** 1Department of Agriculture and Forestry Engineering, Food Technology, College of Agricultural and Forestry Engineering, University of Valladolid, 34004 Valladolid, Spain; 2Ethiopian Institute of Agricultural Research, Addis Ababa 2003, Ethiopia; 3Department of Agroindustrial Engineering, Pacific Littoral Regional University Center, National Autonomous University of Honduras (UNAH), Choluteca 51101, Honduras

**Keywords:** lipase inactivation, free fatty acid, tef grain, microwave treatment

## Abstract

In recent years, many efforts are being made to produce tef-based food for its nutritive and health-promoting advantages. Tef grain is always whole milled because of its tiny grain size and whole flours contain bran (pericarp, aleurone, and germ) where major non-starch lipids could be deposited along with the lipid-degrading enzymes: lipase and lipoxygenase. As lipoxygenase shows little activity in low moisture, the inactivation of lipase is the common objective for most heat treatments to extend the shelf life of flours. In this study, tef flour lipase inactivation kinetics via hydrothermal treatments assisted using microwaves (MW) were studied. The effects of tef flour moisture level (12%, 15%, 20%, and 25%) and MW treatment time (1, 2, 4, 6, and 8 min) on flour lipase activity (LA) and free fatty acid (FFA) content were evaluated. The effects of MW treatment on flour pasting characteristics and the rheological properties of gels prepared from the treated flours were also explored. The inactivation process followed a first-order kinetic response and the apparent rate constant of thermal inactivation increased exponentially with the moisture content of the flour (*M*) according to the equation 0.048·exp (0.073·*M*) (R^2^ = 0.97). The LA of the flours decreased up to 90% under the studied conditions. MW treatment also significantly reduced (up to 20%) the FFA level in the flours. The rheological study confirmed the presence of significant modifications induced by the treatment, as a lateral effect of the flour stabilization process.

## 1. Introduction

In recent years, the market for gluten-free baked products has grown quickly worldwide, more than expected, as a result of the increased incidence of celiac disease [1]. The common raw materials used for developing these products are refined rice and maize because of their high starch content and good technological properties. However, they are low in fiber, protein, and mineral contents and they have bald organoleptic properties [2]. Therefore, technologists and industries are giving more attention to other gluten-free whole grains with high nutritional profiles that can address the needs of people suffering from celiac disease and health-conscious consumers in general. Tef is among these grains for which worldwide interest and acceptance is growing rapidly, because of its attractive nutritional profile characterized by a well-balanced content of all essential amino acids, high content of minerals, polyphenols and dietary fiber [3,4]. A proximate composition study completed on 13 tef grain varieties showed that tef grain contains: moisture, 9.30–11.22% (mean 10.5%); grain protein, 8.7–11.1% (mean 10.4%); ash, 1.99–3.16% (mean 2.45%); crude fat, 2.0–3.0% (mean 2.3%); and crude fiber 2.6–3.8% (mean 3.3%) [3].

Grains can be stored for a long duration without much change in quality as long as their kernels are intact and stored under the recommended conditions. However, during and after the milling of these grains, the lipids undergo many changes due to the action of lipolytic enzymes, which are mainly located in the aleurone, subaleurone, and oil-rich germ regions, which affect the sensory quality of the products [5]. The milling process causes lipase release that degrades the triglycerides, resulting in increased free fatty acids within a short storage duration, affecting the physicochemical properties of the stored flour [6]. Hence, storage of these whole-milled gluten-free grain flours is difficult as they are susceptible to rancidity that leads to the development of an off-flavor. Due to its small size, the tef grain is always whole floured, including its aleurone and subaleurone, together with its relatively large germ compared to other cereal seeds, which contributes to its lipid content that ranges between 2 and 3.7%, of which 84% corresponds to unsaturated fatty acids [7,8]. These factors could make tef flour susceptible to fat deterioration via lipolysis and oxidative rancidity [9], affecting its shelf life and limiting its utilization [10].

Whole flours contain bran (pericarp, aleurone, and germ) where major non-starch lipids could be deposited along with the lipid-degrading enzymes: lipase and lipoxygenase [9]. As lipoxygenase shows little activity in low moisture, the inactivation of lipase is the common objective for most heat treatments to extend the shelf life of flours [11]. Some thermal processing, including hot air [12], hydrothermal treatment [13], and superheated steam [14] have been previously proven to be able to inactivate lipid-degrading enzymes and stabilize wholegrain cereal. However, these treatments are of limited application due to their high energy requirements [15]. Microwaving is rapid and efficient heating for it requires no heat transmission or loss of heat medium (e.g., the steam in steaming). During the microwave (MW) treatment, polar molecules in the sample absorb MW energy and orient themselves with respect to the electric field and the rapid change in their orientation generates heat via molecular friction, resulting in a bulk heating throughout the sample and a faster heating rate compared with conventional heating [16]. According to Alajaji and El-Adawy [17], MW cooking was better than boiling and autoclaving for retaining B-vitamins (Riboflavin, Thiamin, Niacin and Pyridoxine) and minerals in chickpeas. The treatment also helps in sterilizing flour and destroying microorganisms depending on the microwave processing conditions and the moisture level in the sample [18]. In addition, MW heat treatment of varying severity has been able to inactivate lipase and lipoxygenase in cereal bran and germ, which led to a lower change in free fatty acid values during storage [13,19,20,21]. Hence, the concept of microwave treatment to stabilize tef grain flours for their efficient storage could be applied.

In other heat treatments of flour, hydration has been identified as a critical parameter for successful enzyme inactivation [22,23,24]. Taking this into account, a study of the effect of tef flour moisture content on lipase inactivation kinetics during MW radiation seemed to be necessary.

On the other hand, the heat treatment of cereal grains and their flours at varying moisture level causes significant changes in their morphology and physicochemical properties. MW treatment is being applied to flours and starches of gluten-free cereals to modify their properties and expand their range of applications [25,26,27]. Studies showed that MW treatments altered their functional characteristics and changed their pasting properties and the rheological characteristics of the resulting gels.

Therefore, the aim of this work was to establish the inactivation kinetics of lipase in tef flour by means of MW radiation and how it depends on flour moisture content. Flour hydration was varied between 12% and 25%, while MW treatment time was varied between 1 and 8 min. The side effects of flour MW treatment on flour’s important properties, such as the free fatty acid content (FFA), the flour pasting characteristics, and rheological properties of gels prepared from the treated flours, were also tested.

## 2. Results and Discussion

### 2.1. Effect of Microwave Treatment on Lipase Activity

Table 1 presents the treatment conditions used in this work and the identification of the samples used throughout the text. The recorded temperatures in the Teflon^®^ containers ranged between 60 and 121 °C, showing an increasing trend with increasing flour moisture levels and treatment time, as shown in Figure 1. This trend agrees with earlier works [23,27] and this could be due to the fact that water is one of the most important polar molecules, which changes its direction quickly with the effect of microwaves creating kinetic energy that will be converted into heat [28]. Therefore, the extent of changes in the MW-treated tef flours varied accordingly and are discussed below.

The evolution of lipase activities in the MW-treated tef flours, which shows the extent of lipase inactivation achieved, as affected by treatment time and flour moisture level is depicted in Table 2 and Figure 2. MW treatment significantly (*p* > 0.05) reduced the LA from nearly 10% to 94% of the LA in the native flour, depending on the treatment time and the moisture level at which the flours were conditioned. For all moisture levels, MW treatment time had a highly significant effect (*p* < 0.01) on the LA in the resulting flour, and the inactivation level achieved increased with increasing treatment time. The influence of flour moisture level was also important, where the LA reduction increased with increasing flour moisture level. However, the effect of treatment time was more vivid than the flour moisture content (Figure 2). For the same treatment time level, the variability of LA in the treated flours due to flour moisture level started to be significantly visible on flours treated for ≥2 min (Table 2). The reason could be that the 1 min treatment did not allow the temperature in the flour to reach high enough to make a difference compared with the control. The findings corroborate the report by Yadav et al. [13] in pearl millet grains where MW heating decreased lipase activity by up to 93% depending on the moisture level and treatment time, which led to a lower change in free fatty acid value during storage for up to 30 days.

The dependence of lipase inactivation on flour moisture level and MW treatment time was estimated by determining the kinetics of LA reduction by fitting the results to a first-order kinetic model (Equation (1)) as described in Pérez-Quirce et al. [23]:(1)$$A=A_{0}\cdot \;e^{-k\cdot t}\;$$
where *A* = lipase activity (U/g), *t* = treatment time (min), *A*_0_ = lipase activity at *t* = 0 min. The constant *k* is the apparent constant of lipase inactivation (min^−1^), which is dependent on flour moisture level as described in Equation (2):(2)$$k=k_{0}\cdot \;e^{b\cdot M}$$
where *M* = flour moisture content (g H_2_O/100 g flour), *K*_0_ represents the rate of LA inactivation when the flour moisture content is 0% (min^−1^), and *b* is a constant that quantifies the influence of moisture level in the flours on the lipase inactivation rate. This leads to the general equation (Equation (3)) that helps to estimate the effect of both variables, *M* and *t*:(3)$$A=A_{0}\cdot \;e^{-k_{0}\cdot t\cdot e^{\left(b\cdot M\right)}}$$

The estimated value of the *A*_0_ constant after fitting the data set for this specific system was *A*_0_ = 0.62 ± 0.01 U/g, while the values of *k*_0_ and *b* were found to be 0.048 ± 0.007 min^−1^ and 0.073 ± 0.007, respectively. The adjusted correlation coefficient (R^2^) obtained for the regression undertaken was 0.97 and the standard error of the estimate was 0.027 U/g. This underlines how successfully the model explains the changes in the LA of the flours depending on the flour moisture level (12–25%) and the length of the MW treatment time (1–8 min). The estimated value of *A*_0_ by the model is very close to the measured initial LA in the control sample, which was the tef flour before the MW treatment (Table 2), revealing the good agreement of the model with the experimental data. From the correlation equation (Equation (3)) and the estimated values of the constants, in the range of the two variables studied in this experiment, one can observe the exponential effect of MW treatment time and flour moisture level on the lipase inactivation process. In general, it can be confirmed from the inactivation levels achieved with MW treatment, that this treatment may improve the tef flour storage period because lipid hydrolysis to fatty acid by lipase catalysis could be reduced [13].

### 2.2. Free Fatty Acid Content

Free fatty acid (FFA) content in the MW-treated tef flour samples and the control are shown in Table 2. The effects of MW treatment time and flour moisture level were significant, including the interaction of the two factors. For each moisture level, MW treatment up to 2–4 min significantly reduced the FFA contents of the flours, while further treatment showed a tendency to increase the flour FFA content. Such an increasing trend in the flour FFA contents at 6 and 8 min could be due to the hydrolysis of fats favored by the high temperature (>100 °C, Figure 1) and by the residual lipase activity in the flours during the treatment [29]. At an equivalent treatment time, the flour moisture level also had a significant (*p* < 0.05) effect. As can be seen in Table 2, the FFA content of flours treated for ≥4 min showed a decreasing trend with the increasing flour moisture content and this could be due to their lower lipase activity.

### 2.3. Pasting Properties

The MW treatment on the tef flour had a visible effect on the pasting property parameters (Table 2) and the pasting curves (Figure 3). The effect of treatment time was more noticeable than flour moisture level, while their interaction was highly significant (*p* < 0.005). In general, flours treated with long MW treatments showed a higher pasting temperature, while those subjected to short treatments did not show significant differences and this is in agreement with earlier studies [30,31]. The higher pasting temperature of the flour is an indication of its thermal resistance at the start of gelatinization and this fact can be explained by the changes in crystallinity that starch undergoes during MW treatment, and this is more pronounced with the longer treatment duration [25]. According to the same study, the establishment of associations between chains in the amorphous region of starch granules may also play a role.

The remaining pasting parameters that are related to paste viscosity varied significantly showing an initial increase followed by a decreasing trend; the same trend was reported by Kamble et al. [31]. Peak viscosity varied significantly with treatment time showing an initial increase (4–8%) and a subsequent decrease up to nearly 40%, while the variation in flour moisture level did not show a clear trend. Such a decrease in peak viscosity is probably caused by the thermal degradation of amylopectin and amylose in starch granules and could also be correlated with a reduction in their swelling ability and water binding capacity after the longer treatment times [30]. The effect of moisture level is more vivid on trough viscosity as compared to the other pasting parameters. Except for the 12% moisture level, the trough viscosities of the flours treated for up to 6 min were higher than the control, while the trough viscosity of all the 8 min treated samples were lower. Hence, flour MW treatment at a 15–25% moisture level for between 2 and 6 min could probably give ideal tef flour in applications that require high consistency during prolonged cooking [32]. At all moisture levels, the 1 min MW flour treatment resulted in flour with a significantly higher breakdown viscosity, while treatments for more than 2 min led to a sudden deterioration and this probably shows that a long-time MW treatment of tef flour may give flours with low paste viscosity stability under heating and stirring [33]. A noticeable improvement in the FV of MW-treated tef flours was recorded at 15% to 25% flour moisture levels and between 1 and 6 min treatments, depending on the flour moisture levels. Thus, optimizing the moisture level and treatment time accordingly might give tef flour gel products that require a higher stability after cooling. Relatively similar trends were also observed in the SV of the MW-treated tef flours. MW treatment at lower moisture levels and for longer times could be probably effective in slowing such a retrogradation in tef products.

### 2.4. Rheological Properties

The results of the rheological properties of the tef flour gels obtained after MW treatment at a 25% flour moisture content as affected by the treatment time are depicted in Table 3. Only the highest moisture content was evaluated because this value proved to be the most effective for lipase inactivation. In general, the effect of MW treatment time was clearly visible (Figure 4). As compared to the control, the MW treatment up to 6 min significantly (*p* > 0.05) increased τ_max_, while further treatment led to immediate deterioration. A similar trend was reported for buckwheat by Vicente et al. [27], which underlines the importance of optimizing the MW treatment time in order to obtain a firmer and more stable gel. Even though the crossover point values of the gels tend to increase as of the 2 min treatment time, a significant (*p* < 0.05) increase was scored at the 6 min treatment time level, while it suddenly dropped with the 8 min treatment.

The dynamic moduli values obtained from the frequency sweep data revealed all the gels obtained from flours had solid-like behavior because their elastic moduli (G_1_′) were greater than their viscous moduli (G_1_″), making (tan δ)_1_ < 1. The MW treatment of less than 6 min altered the elastic (G_1_′) and viscous (G_1_″) moduli of the resulting gels only moderately. However, the values of the viscoelastic moduli scored by the gels made from the 8 min treated tef flour were significantly lower than the control and the rest of the treatment time levels. This fact was also supported by the significantly higher (tan δ)_1_ value scored by the 8 min-treated flour indicating the less elastic behavior of its gel [34]. The lowest value of the *a* exponent was obtained for the control gel and continuously increased with the increasing treatment time (except in the sample treated for only 1 min) showing an increased effect of frequency on the elastic modulus of gels made from the treated flour. This might indicate the deterioration of the structural stability of the gels with the increasing treatment time [25,26]. Similarly, the values of the *b* exponent started to increase significantly as of the 2 min treatment, indicating the increasing dependency of the G_1_″ of the gels on the frequency. The *c* exponent, which results from the (*b-a*) difference, showed the opposite tendency, which decreased significantly after 4 min of treatment This underlined the dependency of the ratio G_1_″/G_1_′ of the gels on the frequency, and the ratio also decreased with the increasing treatment time.

## 3. Materials and Methods

### 3.1. Tef Flour Preparation

Brown tef obtained from Salutef (Palencia, Spain) was whole floured using a laboratory mill (Perten mill 120, Perten Instruments, Stockholm, Sweden) fitted with a 0.5 mm screen size. The flours were immediately packed in impermeable plastic bags and stored at 4 °C until further analysis and MW treatment.

### 3.2. Microwave Treatment

Adjustment of the moisture contents (MC) of the tef flour was conducted by adding distilled water to reach 15, 20, and 25% from an initial value of 12% on a wet basis. The flours were continuously mixed while adding the water and then stored for 24 h at 4  ±  2 °C with polyethylene bags for moisture equilibration. A total of 75 g of tef flour of each MC level was weighted and placed in a 1000 mL hermetically sealed cylindrical Teflon^®^ container and treated (900 W, 2450 MHz) in a MW oven (SHARP R-342 Osaka, Japan) for 1, 2, 4, 6, and 8 min in cycles of 10 s of radiation and 50 s of rest. During the treatment, the Teflon^®^ containers were continuously rotated (60–70 rpm) to distribute radiation evenly. The temperature in the containers was controlled using Testoterm^®^ temperature strips (TESTO, Madrid, Spain). The treated flours were dried at 35 °C in an incubation chamber (Memmert ICP260, Schwabach, Germany) until their MC reached 12%, and then sieved to <500 µm.

### 3.3. Lipase Activity

The extent of lipase inactivation was determined by measuring the lipase activity in the treated flour samples according to Rose et al. [35], following the slight modification described in Irakli et al. [36]. Briefly, 1 g of defatted tef flour was weighed into each of two test tubes: one blank and one sample. Then, 400 µL of olive oil and 200 µL of distilled water were added to both tubes and mixed. The lipids from the blank were immediately extracted with 5 mL of hexane (2 times). Next, the supernatants after centrifugation at 1000× *g* for 5 min were pooled and evaporated using a rotary vacuum evaporator, and the residue was dissolved again in 4 mL of isooctane. The other test tube (sample) was incubated for 4 h at 40 °C. Following the incubation, lipid extraction was performed as described for the blank, and both extracts were used for the lipase assay. An aliquot of 0.75 mL isooctane extract was mixed with 0.5 mL of 3% (*v*/*v*) pyridine in 5% (*w*/*v*) aqueous cupric acetate. The mixture was shaken for 1 min, centrifuged for 1 min, and the absorbance of the supernatant was read at 715 nm. The extract absorbance was compared with the absorbance of the oleic acid standard solutions (1–10 mM) prepared in isooctane. The lipase activity was expressed as units per g RB (U/g), where 1 U was defined as the micromoles of fatty acid liberated per h, according to Equation (4):(4)$$Lipase\;activity=1000\cdot \frac{\left(4+v\right)\left(A_{s}-A_{B}\right)}{\varepsilon \cdot t\cdot l\cdot s}$$
where *v* = the volume of olive oil added (mL), *A_S_* = the absorbance of the sample after incubation at 715 nm, *A_B_* = the absorbance of blank at 715 nm, ε = the molar absorptivity of oleic acid at 715 nm (M^−1^cm^−1^, *t* = incubation time (h), *l* = path length (1 cm), and *s* = sample weight (g).

### 3.4. Free Fatty Acid Determination

Free fatty acid values were determined using the rapid titrimetric procedure for small grains described in the AACC method 02-02 [37]. The results were calculated as oleic acid equivalent, which was expressed as a percentage of total lipids quantified using the Soxhlet method.

### 3.5. Pasting Properties

Flour pasting properties were determined using a Rapid Visco Analyser (RVA 4500, Perten Instruments, PerkinElmer, Sydney, Australia). The flour (3.5 g) (at 14% basis) was dispersed in distilled water (25 mL) and the slurry was subjected to RVA analysis under programmed heating and cooling conditions. A temperature of 50 °C was applied for 1 min, followed by heating to 95 °C at a rate of 6 °C/min, holding at 95 °C for 5 min, cooling to 50 °C at a rate of 6 °C/min, and holding at 50 °C for 2 min. The paddle speed was set at 160 rpm. The pasting property parameters recorded were peak viscosity (PV), trough viscosity (TV), breakdown viscosity (BV = PV − TV), setback viscosity (SV = FV − TV), final viscosity (FV), and pasting temperature (PT). All samples were analyzed in duplicate.

### 3.6. Rheological Properties

The rheological tests were performed on the control (untreated tef flour) and the MW-treated flour conditioned to a 25% moisture level because the more visible change was anticipated at this flour moisture level. Fresh flour gels obtained from the pasting property analysis described in Section 3.5 were immediately subjected to the assays on a Kinexus Pro+ rheometer (Malvern Instruments Ltd., Malvern, UK) with a serrated parallel plate geometry with a 40 mm diameter. The selected gap between the parallel plate was 1 mm and the tests were conducted at a constant temperature (25 °C). The tef flour gels were placed on the bottom of the plate and left to rest for 5 min to allow relaxation. Strain sweeps were conducted from a 0.1 to 1000% strain at a constant frequency of 1 Hz. Frequency sweeps were carried out from 1 to 10 Hz by applying a constant strain of 1%, which was within the linear viscoelastic region (LVR). The rheological data were analyzed using rSpace rheometry software for Kinexus V1.72 (Malvern Instruments Ltd., UK). From the strain sweeps, the crossover point was determined at the point where G′ = G″ and tan δ = 1, and the maximum stress (τ_max_) applicable within LVR was estimated by locating the stress level at which G′ decreases above 10%, which coincided with the sudden increase in tan δ [38]. Frequency sweep data were fitted to the power-law model, as previously described in Ronda et al. [38]. The coefficients G_1_′, G_1_″, and (tan δ)_1_ represent the elastic and viscous moduli and the loss tangent at 1 Hz, respectively, obtained by fitting the power law to the frequency sweep data ranging from 1 to 10 Hz. The values *a*, *b*, and *c* are the exponents of the corresponding potential equations (power-law model) and quantify the dependence of the dynamic moduli on the oscillation frequency. All tests were completed in duplicate.

### 3.7. Statistical Analysis

Statistical analysis was conducted using Statgraphics Centurion 18 (Bitstream, Cambridge, MN, USA). The least significant difference (LSD) analysis of variance (ANOVA) was used to assess the significant differences (*p* < 0.05) between samples.

## 4. Conclusions

The result of the MW treatment of tef flour can be considered a promising alternative for reducing lipase activity in tef flour. The lipase inactivation process followed a first-order kinetic response (R^2^ = 0.97). The apparent rate constant of thermal inactivation increased exponentially with the moisture content of the flour and treatment time. However, despite the effectiveness of a longer treatment time and higher moisture level, caution should be taken due to the fact that severe treatments, above 6 min of MW radiation, may lead to deleterious effects on the flour by generating higher FFA in the flours and lowering the strength and consistency of the gels obtained from the flours.

## Figures and Tables

**Figure 1 molecules-28-02298-f001:**
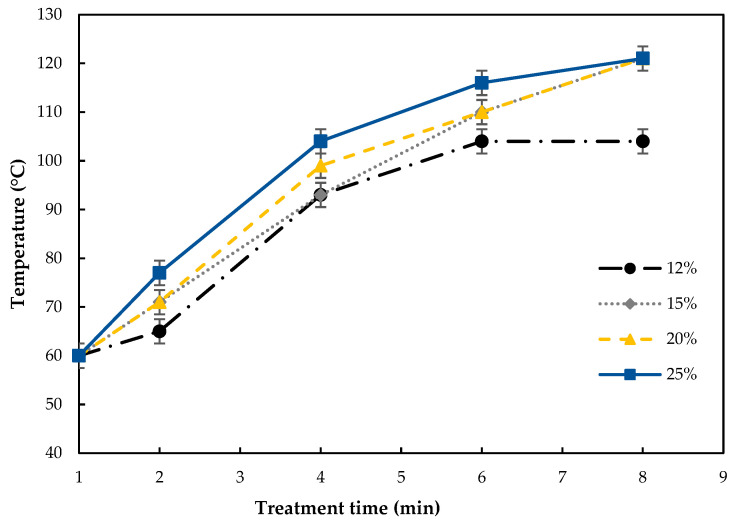
Evolution of the temperature of tef flour during the treatment at 12%, 15%, 20%, and 25% moisture content.

**Figure 2 molecules-28-02298-f002:**
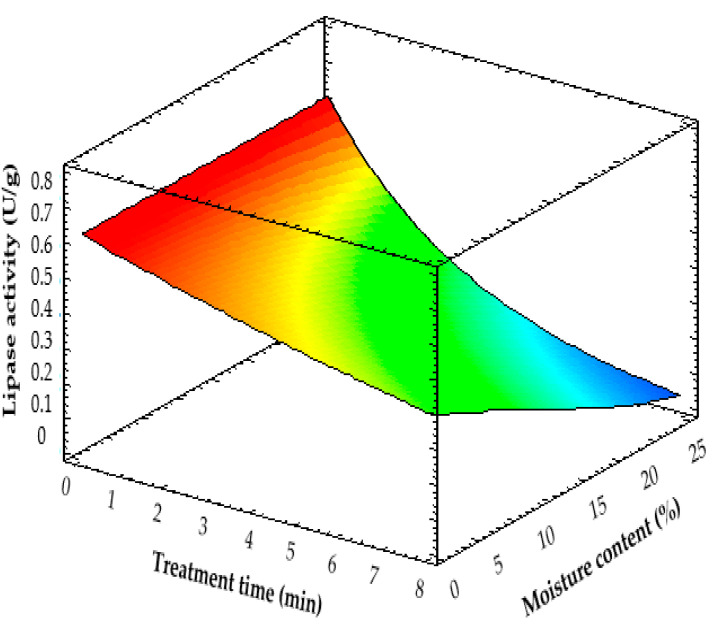
Kinetics of lipase inactivation in tef flour via microwave treatment depending on treatment time (*t*) and moisture content (*M*). Surface corresponds to the fitting equation: 0.62·exp(ࢤ0.048·*t*·exp(0.073·*M*)).

**Figure 3 molecules-28-02298-f003:**
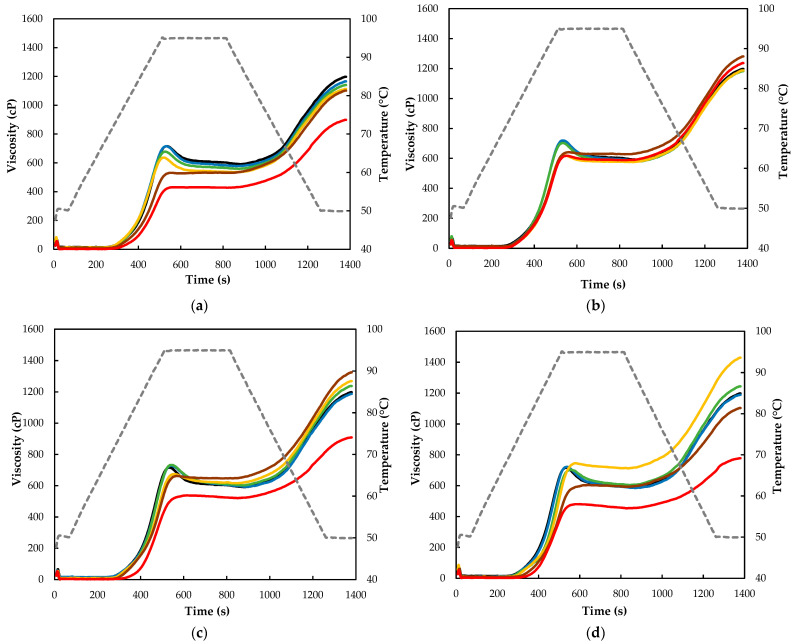
Pasting profiles of control and MW-treated brown tef flour at 12 (**a**), 15 (**b**), 20 (**c**), and 25% moisture content (**d**). tef flour native (control). Black, blue, green, yellow, brown, and red lines correspond to untreated, 1 min, 2 min, 4 min, 6 min, and 8 min treated samples, respectively, and grey discontinuous lines correspond to the temperature profile.

**Figure 4 molecules-28-02298-f004:**
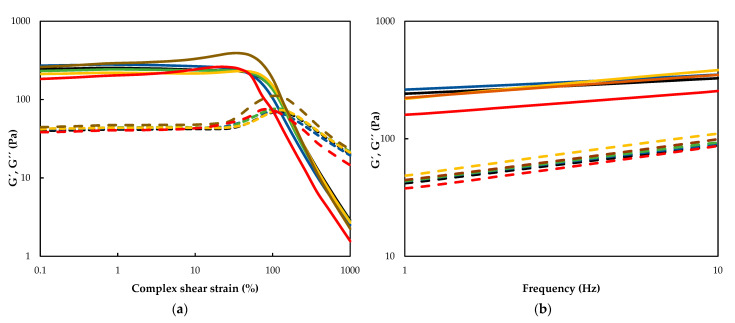
Strain sweeps (**a**) and frequency sweeps (**b**) of the analyzed flour gel samples. Black, blue, green, yellow, brown and red lines correspond to untreated, 1 min, 2 min, 4 min, 6 min, and 8 min treated samples, respectively. G′ is represented by the solid line, while G″ is represented by the dotted line.

**Table 1 molecules-28-02298-t001:** Microwave treatment conditions and identification of samples.

Sample	Moisture Content (%)	Treatment Time (min)
BT (Control)	12	-
BT-12-1	12	1
BT-12-2	12	2
BT-12-4	12	4
BT-12-6	12	6
BT-12-8	12	8
BT-15-1	15	1
BT-15-2	15	2
BT-15-4	15	4
BT-15-6	15	6
BT-15-8	15	8
BT-20-1	20	1
BT-20-2	20	2
BT-20-4	20	4
BT-20-6	20	6
BT-20-8	20	8
BT-25-1	25	1
BT-25-2	25	2
BT-25-4	25	4
BT-25-6	25	6
BT-25-8	25	8

BT (Control) refers to untreated brown tef flour.

**Table 2 molecules-28-02298-t002:** Lipase activity, free fatty acids, and pasting properties of samples analyzed.

Sample	Moisture Content (%)	Time (min)	Lipase Activity (U/g)	Free Fatty Acids (%)	PT (°C)	PV (cP)	TV (cP)	BV (cP)	FV (cP)	SV (cP)
Control			0.62 d	6.25 e	71.1 a	708 e	588 d	120 c	1190 c	602 c
BT12-1	12	1	0.56 cA	5.16 aA	70.9 aA	728 eA	585 dA	144 dA	1184 cA	600 cA
BT12-2		2	0.50 cB	5.53 bB	70.9 aA	673 dA	556 cA	117 cAB	1134 bA	578 bA
BT12-4		4	0.41 bD	5.75 cB	71.3 aA	644 cB	542 bcA	102 bD	1125 bA	583 bcA
BT12-6		6	0.26 aC	6.04 dB	72.5 bA	535 bA	535 bA	2 aA	1109 bA	575 bB
BT12-8		8	0.28 aC	8.07 fC	74.2 cB	432 aA	429 aA	3 aA	902 aB	473 aC
SE			0.02	0.08	0.2	7	4	3	10	6
Control			0.62 f	6.25 c	71.1 a	708 c	588 a	120 d	1190 a	602 a
BT15-1	15	1	0.50 eA	5.8 aC	71.5 abB	735 cA	595 aA	140 eA	1207 abA	612 aA
BT15-2		2	0.39 dA	5.6 aB	71.8 bB	715 cB	583 aB	133 deC	1192 abB	610 aB
BT15-4		4	0.34 cC	6.0 bC	72.3 cBC	615 aA	573 aB	43 cB	1185 aB	613 aB
BT15-6		6	0.27 bC	6.3 cB	72.5 cA	658 bC	643 bC	15 aC	1310 cB	667 bC
BT15-8		8	0.04 aA	7.2 dB	72.7 cA	622 abD	594 aD	29 bC	1247 bC	654 bD
SE			0.01	0.1	0.1	11	8	4	16	8
Control			0.62 f	6.25 c	71.1 a	708 cd	588 b	120 c	1190 b	602 b
BT20-1	20	1	0.53 eA	5.2 aA	71.7 bcB	755 eA	610 bA	146 dA	1222 bcA	612 bA
BT20-2		2	0.39 dA	5.1 aA	71.3 abAB	725 deB	598 bBC	127 cBC	1227 bcC	630 bcC
BT20-4		4	0.29 cB	5.4 bA	71.9 cB	673 bcC	615 bC	58 bC	1269 cC	654 cdC
BT20-6		6	0.20 bB	5.6 bA	72.7 dA	666 bC	651 cC	15 aC	1332 dB	682 dC
BT20-8		8	0.09 aB	7.2 dB	77.4 eC	542 aC	527 aC	15 aB	920 aB	393 aB
SE			0.01	0.1	0.1	12	8	5	16	8
Control			0.62 e	6.25 b	71.13 a	708 c	588 b	120 d	1190 c	602 c
BT25-1	25	1	0.52 dA	5.4 aB	71.50 bA	735 deA	592 bA	143 eA	1205 cA	613 cA
BT25-2		2	0.37 cA	5.3 aAB	71.48 bAB	713 cdB	611 cC	102 cA	1252 dC	641 dC
BT25-4		4	0.15 bA	5.3 aA	72.70 cC	748 eD	715 dD	33 bA	1437 eD	722 eD
BT25-6		6	0.07 aA	5.5 aA	75.25 dB	608 bB	599 bcB	9 aB	1116 bA	517 bA
BT25-8		8	0.07 aB	6.2 bA	77.63 eC	486 aB	457 aB	29 bC	774 aA	317 aA
SE			0.02	0.2	0.08	7	5	3	10	6
Analysis of variance and significance (*p*-values)										
Moisture content (F1)			ns	*	ns	ns	*	ns	ns	ns
Treatment time (F2)			***	***	***	***	***	***	***	***
(F1) × (F2)			***	***	***	***	***	***	***	***

PT = Pasting Temperature. PV = Peak Viscosity. TV = Trough Viscosity. BV = Breakdown Viscosity. FV = Final Viscosity. SV = Setback Viscosity. SE: Pooled standard error from ANOVA. The different letters in the corresponding column within each studied factor indicate statistically significant differences between means at *p* < 0.05. Lowercase letters are used to compare the effect of treatment time within the same moisture content and capital letters compare the effect of moisture content within the same treatment time. Analysis of variance and significance: *** *p* < 0.001. * *p* < 0.05. ns: not significant.

**Table 3 molecules-28-02298-t003:** Rheological properties of MW-treated samples at 25% moisture content.

Sample	G_1_′ (Pa)	*a*	G_1_” (Pa)	*b*	(Tan δ)_1_	*c*	τ_max_ (Pa)	Crossover (Pa)
Control	239 c	0.133 a	41.3 b	0.321 a	0.173 ab	0.188 d	116 b	144 b
BT25-1	263 d	0.127 a	43.3 bc	0.318 a	0.165 a	0.191 d	136 c	141 b
BT25-2	232 c	0.149 b	43.7 c	0.331 b	0.189 bc	0.182 cd	121 bc	148 b
BT25-4	209 b	0.168 c	44.1 c	0.341 c	0.211 d	0.173 bc	162 d	164 b
BT25-6	223 bc	0.193 d	44.1 c	0.348 d	0.197 cd	0.155 a	202 e	208 c
BT25-8	155 a	0.201 d	35.8 a	0.365 e	0.231 e	0.163 ab	93 a	103 a
SE	6	0.004	0.6	0.001	0.005	0.003	5	7

G_1_′ (elastic modulus), G_1_″ (viscous modulus) and (Tan δ)_1_ (loss tangent) are the coefficients obtained from fitting the frequency sweep data to the power law model and represent the moduli and loss tangent values at a frequency of 1 Hz. The *a*, *b*, and *c* exponents quantify the dependence degree of dynamic moduli and the loss tangent with the oscillation frequency. τ_max_ represents the maximum stress tolerated by the sample in the LVR. SE: Pooled standard error from ANOVA. Different letters in the same column of each individual sample indicate statistically significant differences between means at *p* < 0.05.

## Data Availability

The data presented in this study are available on request from the corresponding author.

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
