# Peer review of "Lipase Inactivation Kinetics of Tef Flour with Microwave Radiation and Impact on the Rheological Properties of the Gels Made from Treated Flour"

_molecules, 2023, doi:10.3390/molecules28052298_

Round 1

Reviewer 1 Report

The manuscript is fine, altough the methodology is rather simple it could give some more information regarding the inactivation of kinetics of tef flour and its application, 

However the information about the applied model for the free fatty acid content is scarce and should be improved. 

Include the p-val of each parameter and the one of the model. Include the lack of fit test result (p-value) and the adjusted R2. All this parameters must be adequate.

Reviewer 2 Report

The article is devoted to the effect of microwave radiation on flours from different types of teff in order to eliminate lipases in the grain. The influence of the microwave effect on some qualities of the flour was monitored. The paper contains important data on a new approach to flour processing and deserves publication.

I have the following comments and questions that were not discussed in the presentation.

1) Can the microwave effect also affect other enzymes in the composition of the grain? Of particular importance is whether vitamins such as riboflavin, thiamin, niacin and vitamin A, which are some of the most beneficial ingredients of teff flour, are broken down during the microwave irradiation process.

2) Are there any data on whether microwave irradiation affects the taste qualities of the flour?

3) I think that the introduction could discuss in more detail the composition of the grains themselves, and also the differences between the types of teff that are included in the study, and in general.

4) Do you think that microwaves sterilize flour and kill any harmful microorganisms?

4) I have some minor remarks:

- on page 2, lines 62-63 - the sentence is for correction.

- The mention of figure 2 in the text is too far from the figure itself.
